# Automatic Grouping for Efficient Cooperative Multi-Agent Reinforcement Learning

**Yifan Zang**[1,2], **Jinmin He**[1,2], **Kai Li**[1,2*], **Haobo Fu**[3], **Qiang Fu**[3], **Junliang Xing**[4], **Jian Cheng**[1,2*]

[1]Institute of Automation, Chinese Academy of Sciences
[2]School of Artificial Intelligence, University of Chinese Academy of Sciences
[3]Tencent AI Lab, [4]Tsinghua University
{zangyifan2019,hejinmin2021,kai.li}@ia.ac.cn,
{haobofu,leonfu}@tencent.com,jlxing@tsinghua.edu.cn,jcheng@nlpr.ia.ac.cn

## Abstract

Grouping is ubiquitous in natural systems and is essential for promoting efficiency in team coordination. This paper proposes a novel formulation of Group-oriented Multi-Agent Reinforcement Learning (GoMARL), which learns automatic grouping without domain knowledge for efficient cooperation. In contrast to existing approaches that attempt to directly learn the complex relationship between the joint action-values and individual utilities, we empower subgroups as a bridge to model the connection between small sets of agents and encourage cooperation among them, thereby improving the learning efficiency of the whole team. In particular, we factorize the joint action-values as a combination of group-wise values, which guide agents to improve their policies in a fine-grained fashion. We present an automatic grouping mechanism to generate dynamic groups and group action-values. We further introduce a hierarchical control for policy learning that drives the agents in the same group to specialize in similar policies and possess diverse strategies for various groups. Experiments on the StarCraft II micromanagement tasks and Google Research Football scenarios verify our method's effectiveness. Extensive component studies show how grouping works and enhances performance.

## 1 Introduction

Cooperative multi-agent reinforcement learning (MARL) aims to coordinate multiple agents' actions through shared team rewards and has become a helpful tool for solving multi-agent decision-making problems [47, 9, 44]. Learning centralized policies to address this problem conditions on the full state, which is usually unavailable during execution due to partial observability or communication constraints [4, 3]. An alternative paradigm is to learn decentralized policies based on local observations [36, 31]. However, simultaneous exploration suffers from non-stationarity that may cause unstable learning [7, 50]. The centralized training with decentralized execution (CTDE) inherits the advantages of these two paradigms and learns decentralized policies in a centralized fashion [23, 15].

Value Function Factorization (VFF) under the CTDE paradigm is a popular approach to MARL, where a centralized value function is learned from global rewards and factorized into local values to train decentralized policies. In recent years, value factorization methods have been widely proposed. Existing methods usually adopt the flat VFF scheme [24], which directly estimates the joint action-value from local utilities. These methods achieve remarkable performances from various perspectives, such as enhancing the expressiveness of the mixing network [27, 33, 38, 37] and encouraging exploration [21, 19]. However, learning efficient cooperation by directly estimating the joint action-values from individual utilities is exceptionally difficult. Phan et al.[24] illustrate that the flat VFF

---

*Corresponding authors.

scheme leads to a performance bottleneck, where it gets challenging to provide sufficiently informative training signals for each agent. Although the state information provides complete knowledge, it is burdensome for agents to extract effective guidance that facilitates cooperative policy learning.

Research in natural systems [13, 43] and multi-agent systems [24] has validated grouping as a means to promote efficient collaboration. Dividing the team into smaller sets allows for fine-grained learning and opens up further opportunities to integrate informative group-wise learning signals. However, formulating a general division criterion without domain knowledge is still a matter of interest to the community. Most previous works are proposed for well-structured tasks and typically predefine specific responsibilities or forms of task decomposition [1, 18, 24]. They require apriori settings that are potentially unavailable in practice and discourage methods' transferring to diverse environments.

This paper proposes a novel formulation of Group-oriented MARL (GoMARL), which learns automatic grouping without domain knowledge for efficient cooperation. GoMARL holds a dual-hierarchical value factorization and learns dynamic groups with a "select-and-kick-out" scheme. Concretely, GoMARL continuously selects agents unsuitable for their current groups based on the learning weights of the decomposition from the group-wise value to local utilities and kicks them out to reorganize the group division. Furthermore, GoMARL transforms various informative training signals, including individual group-related information, group state, and global state, into network weights, which extracts effective guidance for policy improvement and enables flexible adaptation to the dynamic changes in the number of subgroups and the number of agents per group.

The advantages of GoMARL lie in two aspects. Firstly, group information offers richer knowledge for group-wise value factorization and provides more direct guidance for efficient learning. Our value factorization with a dual hierarchy is based on a more focused and compact input representation. Unlike existing methods that learn value factorization with no information or only with the global state which is complete but hard to extract efficient guidance, GoMARL proposes a fine-grained learning scheme to integrate information from the group perspective into the policy gradient, promoting intra-group coordination with the group state and facilitating inter-group cooperation with the global state. Secondly, GoMARL extracts individual group-related information to provide informative signals for policy diversity. The agents condition their behaviors on their individual group-related information embedded by a shared encoder, which is trained following specialization guidance, *i.e.*, imposing similarity within a group and diversity among groups. In this way, GoMARL synergizes subgroups with policy styles, proposing a parameter-sharing mechanism for specialized policies.

We test our method on a challenging set of StarCraft II micromanagement tasks [29] and Google Research Football scenarios [16]. GoMARL achieves superior performance with greater efficiency compared with notable baseline methods. We also conduct detailed component analyses and ablation studies to give insights into how each module works and enhances learning efficiency.

## 2 Related Work

Value function factorization under the CTDE [23] paradigm is a popular approach to MARL. Most existing methods learn flat value factorization by treating agents as independent factors and directly estimating the joint action-value from local utilities. The earlier work, VDN [35], learns a linear decomposition into a sum of local utilities used for greedy action selection. QMIX [27] learns a non-linear mixing network with the global state and enlarges the functions the mixing network can represent, but it still faces the monotonicity constraint. QTRAN [33] further improves the expressivity by proposing the Individual-Global-Max (IGM) principle between individual utilities and the global action-value. Subsequent works follow the IGM principle and further boost performance by encouraging exploration [21, 5], enhancing the expressiveness of the mixing network [38], preventing sub-optimal convergences [26, 37], and integrating functional modules [42, 46, 45, 51, 34, 49].

This paper explores automatic group division for multi-agent teams to realize group-wise learning. Early related works [30, 28, 20, 18] predefine specific responsibilities to each agent based on goal, visibility, capability, or by search. Some other efforts achieve implicit grouping by task allocation [32, 17, 10]. These methods only address tasks with a clear structure and require domain knowledge or apriori settings. Another class of approaches focuses on individuality [41, 19, 14] or role learning [39, 40]. Among them, ROMA [39] learns dynamic roles that depend on the context agents observe. RODE [40] decomposes the joint action spaces and integrates the action effects into the role policies to boost learning. A similar work, VAST [24], also studies the impact of subgroups

on value factorization but still requires a priori of group number. Our method, GoMARL, does not rely on domain knowledge and gradually adjusts the group division according to the learned factorization weights. Based on the dynamic group division, GoMARL models several group-related signals to learn specialized policies and promote efficient team cooperation.

## 3 Preliminaries

**Dec-POMDP.** This paper focuses on cooperative tasks with $n$ agents $\mathcal{A} = \{a_1, ..., a_n\}$ as a Dec-POMDP [22] defined by a tuple $G = \langle S, U, P, r, Z, O, n, \gamma \rangle$. The environment has a global *state* $s \in S$. Each agent $a$ chooses an *action* $u_t^a$ from its action space $U_a$ at timestep $t$ and forms a joint action $\mathbf{u}_t \in (U_1 \times ... \times U_n) \equiv U^n$ that induces a transition according to the *state transition distribution* $P(s_{t+1}|s_t, \mathbf{u}_t) : S \times U^n \times S \to [0, 1]$. $r(s, \mathbf{u}) : S \times U^n \to \mathbb{R}$ is the *reward* function yielding a shared reward, and $\gamma \in [0, 1)$ is the discount factor. We consider partially observable scenarios in which agent $a$ acquires its local *observation* $z^a \in Z$ drawn from $O(s_t, a) : S \times \mathcal{A} \to Z$ and has an *action-observation history* $\tau^a \in T \equiv (U \times Z)^*$ on which it conditions a *policy* $\pi^a(u^a|\tau^a) : T \times U \to [0, 1]$.

**Value Function Factorization and the IGM Principle.** We consider cooperative MARL with the CTDE paradigm, which has been a significant focus in recent efforts [2, 35, 27, 12, 21]. A majority of methods achieve CTDE through flat value function factorization [24], *i.e.*, factoring action-value functions into combinations of per-agent utilities. The individual utility only depends on the local history of actions and observations, allowing agents to maximize their local utility functions independently under the Individual-Global-Max (IGM) Principle [33]: $\arg\max_{\mathbf{u}} Q^{tot}(\boldsymbol{\tau}, \mathbf{u}) = \{\arg\max_{u^1} Q^1(\tau^1, u^1), \cdots, \arg\max_{u^n} Q^n(\tau^n, u^n)\}$. Among these attempts, the representative deep MARL approach QMIX [27] improves the simple summation of individual utilities [35] by introducing a more expressive factorization: $Q^{tot} = f(Q^1(\tau^1, u^1; \theta_Q), \cdots, Q^n(\tau^n, u^n; \theta_Q); \theta_f)$, where $\theta_f$ denotes the parameters of the monotonic mixing function generated by a hypernetwork [6].

## 4 Group-oriented Multi-Agent Reinforcement Learning

**Definition 1. (Individual and Group)** Given a cooperative task with $n$ agents $\mathcal{A} = \{a_1, ..., a_n\}$, we have a set of groups $\mathcal{G} = \{g_1, ..., g_m\}$, $1 \leq m \leq n$. Each group $g_j$ contains $n_j$ $(1 \leq n_j \leq n)$ different agents, $g_j = \{a_{j_1}, ..., a_{j_{n_j}}\} \subseteq \mathcal{A}$, where $\bigcup_j g_j = \mathcal{A}$, $g_j \cap g_k = \varnothing$ for $j, k \in \{1, 2, \ldots, m\}$ and $j \neq k$. The *superscript* describes the variable owner, *e.g.*, $u^{j_i}$ is the action of the *i-th* agent $a_{j_i}$ in group $g_j$. We denote joint quantities in bold and joint quantities over agents other than a given agent $a$ with the superscript $-a$, *e.g.*, $\mathbf{u}^{-j_i}$ is the joint action of agents in group $g_j$ other than agent $a_{j_i}$.

GoMARL decomposes the global action-value $Q^{tot}$ into group-wise values $Q^g$ and trains agents by groups in a fine-grained manner. Figure 1 illustrates the overview of the learning framework. It consists of an automatic grouping module, specialized agent networks generating local utilities $Q^i$, and a mixing network among groups. In Section 4.1, we introduce the automatic grouping module. It progressively divides the team into dynamic groups as training proceeds. Based on the dynamic group division, we propose specialized agent networks that achieve similarity within each group and diversity among groups to generate local utilities $Q^i$ in Section 4.2. Section 4.3 presents the overall training framework to detail the estimation of the group-wise action-values and the global action-values.

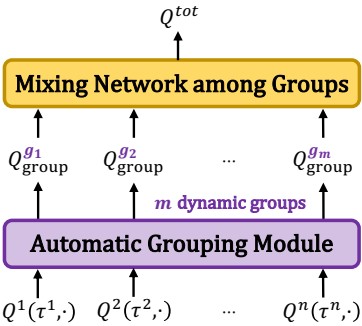

Figure 1: Overview of GoMARL.

### 4.1 Automatic Grouping Mechanism

The automatic grouping mechanism aims to learn a mapping relationship $f_g : \mathcal{A} \mapsto \mathcal{G}$. The key idea is to divide the team into dynamic groups in an end-to-end fashion by maximizing the expected global return $Q_{\mathcal{G}}^{tot}(s_t, \mathbf{u}_t) = \mathbb{E}_{s_{t+1:\infty}, \mathbf{u}_{t+1:\infty}} \left[ \sum_{k=0}^{\infty} \gamma^k r_{t+k} | s_t, \mathbf{u}_t; \mathcal{G} \right]$. Value function factorization approaches represent the joint action-value as an aggregation of individual utilities, *i.e.*, a weighted sum of $Q^i$ and biases. We follow this setting and represent the group-wise value $Q^g$ as an aggregation of the individual value $Q^i$. Intuitively, as illustrated in the right side of Figure 2, if the learned mixing

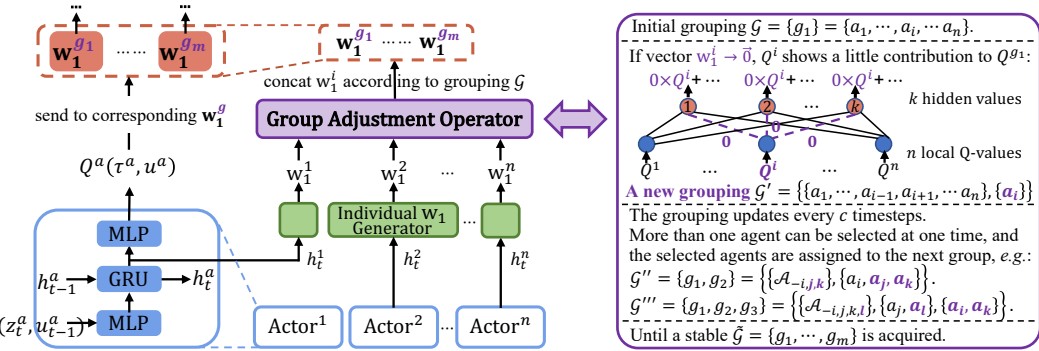

Figure 2: The left side illustrates the schematic diagram of the grouping mechanism. The group adjustment operator concatenates the weights $\mathrm{w}_1^i$ of agents in the same group to form the group-wise weights $\mathrm{w}_1^g$. The right side gives an example to show how grouping $\mathcal{G}$ changes during training.

weight of $Q^i$ is small enough, then $Q^i$ contributes a little to its group-wise value $Q^g$. In other words, when an agent $a_{j_i}$ takes action $u^{j_i}$ but does not affect its group-wise value $Q^{g_j}$, it indicates that agent $a_{j_i}$ does not belong to group $g_j$ anymore and needs group adjustment.

We factorize the group value $Q^g$ into individual utilities $Q^i$ with group-wise mixing weights $\mathrm{w}_1$, *i.e.*, $Q^{g_j} = f(Q^{j_1}(\tau^{j_1}, u^{j_1}), \cdots, Q^{j_{n_j}}(\tau^{j_{n_j}}, u^{j_{n_j}}); \mathbf{w}_1^{g_j})$, where $\mathbf{w}_1^{g_j}$ denotes group $g_j$'s mixing parameters generated by individual $\mathrm{w}_1$ generators $f_{\mathrm{w}_1}^i(\cdot; \theta_{\mathrm{w}_1}^i) : \tau^i \to \mathrm{w}_1^i$ that map each agent $a_i$'s history information $\tau^i$ to a $k$-dimensional weight vector. A regularization on this $\mathrm{w}_1$ generator $f_{\mathrm{w}_1}^i$ drives the automatic grouping mechanism. Specifically, GoMARL "selects and kicks out" agents whose individual utilities hold small mixing weights and contribute a little to their group-wise $Q$ values; thus, a sparsity regularization on $\mathrm{w}_1$ is implemented to select those agents with small group-value mixing weights. The $\mathrm{w}_1$ generators $f_{\mathrm{w}_1}^i$ are trained by minimizing the following loss function:

$$\mathcal{L}_g\left(\theta_{\mathrm{w}_1}\right) = \mathbb{E}_{(\mathbf{z}, \mathbf{u}, r, \mathbf{z}') \sim \mathcal{B}} \sum_i \left( \| f_{\mathrm{w}_1}^i(\tau^i(z^i, u^i); \theta_{\mathrm{w}_1}^i) \|_{l_1} \right), \tag{1}$$

where $\mathcal{B}$ is the replay buffer, and $\| \cdot \|_{l_1}$ stands for the $l_1$-norm penalty.

Figure 2 illustrates the schematic diagram of the grouping mechanism. Initially, all the agents belong to the same group, and the grouping $\mathcal{G}$ is adjusted as training proceeds. The grouping shifts every $c$ timesteps, and each selected agent is assigned to the following group (a new one for agents in the last group) until it properly contributes to where it belongs. When adjusting groups, the $\mathrm{w}_1$ of all agents are sent to a group adjustment operator $\mathcal{O}_g$. According to the adjusted grouping $\mathcal{G}$, $\mathcal{O}_g : \{\mathrm{w}_1^1, \cdots, \mathrm{w}_1^n\} \xrightarrow{\mathcal{G}} \{\mathbf{w}_1^{g_1}, \cdots, \mathbf{w}_1^{g_m}\}$ concatenates the $\mathrm{w}_1^i$ of agents in the same group to form a set of group-wise $\mathbf{w}_1^g$ to generate group action-value. This implementation flexibly adapts to the grouping dynamics since each $\mathrm{w}_1^i$ is tied to agent $a_i$ by the individual $\mathrm{w}_1$ generator $f_{\mathrm{w}_1}^i$. No matter which group $g_j$ agent $a_i$ belongs to, $Q^i$ can engage in the estimation of $Q^{g_j}$ through $\mathrm{w}_1^i \in \mathbf{w}_1^{g_j}$.

In practice, the weights' shrinkage to zero is infeasible. It is also challenging to determine a fixed threshold for groups of various sizes in diverse environments. We empirically utilize seventy percent of each group's average weight to assess whether an agent fits its current group. Extensive experiments and component studies in Section 5 confirm the universality of this configuration.

## 4.2 Specialized Agent Network for Decentralized Execution

Group status describes the cooperation among agents within each group and is essential to local utility generation. Integrating group-wise information $e$ into the decision-making process enables consideration of cooperative behaviors. In this section, we introduce GoMARL's agent network to generate local action-value functions with individual group-related information $e$ embedded.

As shown in Figure 3, we construct a group-related info encoder $f_e(\cdot; \theta_e)$ to embed agents' hidden states. To achieve a group-related view, we train the encoder network as an extractor, where the extracted agent info $e$ of agents from the same group should be similar. To avoid all agents' $e^i$

collapsing to be alike, the regularizer also encourages diversity between agents from different groups. Formally, we minimize the following similarity-diversity objective to train the encoder $f_e$:

$$\mathcal{L}_{SD}\left(\theta_e\right) = \mathbb{E}_{\mathcal{B}}\Big(\sum_{i \neq j} I(i,j) \cdot \text{cosine}\big(f_e(h^i; \theta_e), f_e(h^j; \theta_e)\big)\Big),$$

$$\text{where } I(i,j) = \begin{cases} -1, & a_i, a_j \in g_k. \\ 1, & a_i \in g_k, a_j \in g_l, k \neq l. \end{cases} \tag{2}$$

The encoder trained by the SD-loss extracts agent info $e^i$ that is recognizable to agents' groups. This group-related information is then fed into a decoder $f_d\left(\cdot; \theta_d\right)$ to generate the parameters of the agent network's upper MLP. In this way, each agent $a_i$ conditions its behaviors on $e^i$ with group-related information embedded, promoting potential cooperation within each group during decentralized execution. The decoder hypernetwork $f_d$ is trained by the TD-loss introduced in Section 4.3.

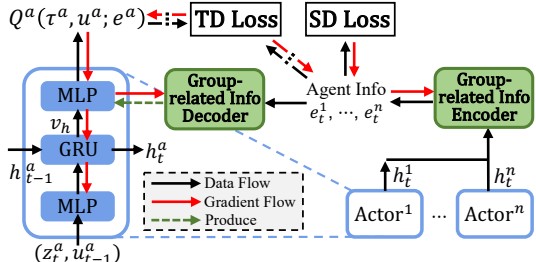

Figure 3: The specialized agent network.

The proposed actor network has two merits. Firstly, it enables diversified policies while sharing all the parameters. Policy decentralization with shared parameters is widely utilized to improve scalability and learning efficiency. However, agents tend to behave similarly when sharing parameters, preventing effective exploration and complex cooperative policies. It is also undesirable to entirely forgo shared parameters in pursuit of diversity since proper sharing accelerates learning. Our method hybridizes the efficiency of parameter-sharing and the policy diversity needed for complex collaboration. Secondly, the decoder hypernetwork $f_d$ integrates the extracted agent info $e$ into the policy gradients, providing informative group-related information to enrich local utilities and promote intra-group cooperation. Concretely, the partial derivative for updating the parameters $\theta_h$ of the GRU and the bottom MLP is:

$$\frac{\partial Q^{tot}}{\partial \theta_h} = \frac{\partial Q^{tot}}{\partial Q^a}\frac{\partial Q^a}{\partial \theta_h} = \frac{\partial Q^{tot}}{\partial Q^a}\frac{\partial Q^a}{\partial v_h^a}\frac{\partial v_h^a}{\partial \theta_h} = f_d(e^a) \cdot \frac{\partial Q^{tot}}{\partial Q^a}\frac{\partial v_h^a}{\partial \theta_h}, \tag{3}$$

where $v_h$ is the representation after the GRU illustrated in the left side of Figure 3. Eqn.(3) shows that $e^a$ is deeply involved in the policy updating of agent $a$, providing richer group knowledge to generate local utilities and facilitating group-related guidance for cooperative decentralized execution.

## 4.3 Overall Learning Framework

We next introduce GoMARL's overall learning framework. It contains two mixing networks that generate the group-wise $Q^g$ and the global $Q^{tot}$, respectively. As for the group mixing network, although $\mathbf{w}_1^g$ enables a weighted mixing of local utilities of agents in group $g$, the naive mixture of $\sum_{a \in g} \mathbf{w}_1^a Q^a + b$ lacks the guidance of group status to reflect the group action-value in a specific *group-wise state*. Therefore, we build the group mixing network with a two-layer structure that also embeds the group-wise state

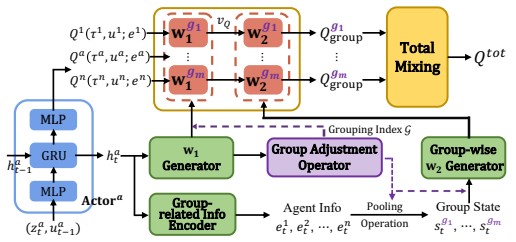

Figure 4: Overall learning framework of GoMARL.

into the weights $\mathbf{w}_2^g$ to generate $Q^g$. In particular, group $g_j$'s group-wise state $s^{g_j}$ is a fusion of the agent info $e^i$ for all $a_i \in g_j$. To cohesively summarize the group state based on the agent info of all agents in group $g_j$, we apply a pooling operation [25] over each dimension of the agent info $e^i$ to generate the group-wise state $s^{g_j}$ describing the current group status. The pooling operation also ensures adaptability to the dynamic group size (*i.e.*, the number of agents per group). We build a group-wise $\mathbf{w}_2$ generator $f_{w_2}(s^g)$ to map the fused group state into $\mathbf{w}_2^g$. Similar to the group-related info encoder $f_e$, the $\mathbf{w}_2$ generator integrates the group state $s^g$ into the policy gradient: $\frac{\partial Q^g}{\partial Q^a} = \frac{\partial Q^g}{\partial v_Q^a}\frac{\partial v_Q^a}{\partial Q^a} = f_{w_2}(s^g) \cdot \frac{\partial v_Q^a}{\partial Q^a}$, providing group status information for $Q^g$ generation that facilitates efficient *intra-group cooperation*. $v_Q$ is the representation after $\mathbf{w}_1^g$ marked in Figure 4. The

two-layer mixing structure of the group mixing network estimates the group action-value $Q^g$ by $\mathbf{w}_1$ which decides the group division and $\mathbf{w}_2$ which carries group status information.

GoMARL estimates the global action-value $Q^{tot}$ by mixing all the $Q^g$ in a similar fashion. Concretely, the two layers of the total mixing network are generated by two hypernetworks, respectively taking group states $s^g$ and the full state $s$ as inputs. Likewise, $s^g$ and $s$ are deeply involved in the gradients, promoting *inter-group cooperation*. The architecture of the total mixing network is akin to the group mixing network and is omitted in Figure 4 for brevity. The TD-loss of the estimated $Q^{tot}$ is:

$$\mathcal{L}_{TD}(\theta) = \mathbb{E}_{\mathcal{B}}\left[\left(r + \gamma \max_{\mathbf{u}'} \bar{Q}^{tot}\left(s', \mathbf{u}'\right) - Q^{tot}\left(s, \mathbf{u}\right)\right)^2\right], \tag{4}$$

where $\bar{Q}^{tot}$ is a target network with periodic updates. The overall learning objective of GoMARL is:

$$\mathcal{L}(\theta) = \mathcal{L}_{TD}(\theta) + \lambda_g \mathcal{L}_g(\theta_{\mathbf{w}_1}) + \lambda_{SD}\mathcal{L}_{SD}(\theta_e), \tag{5}$$

where $\theta = (\theta_h, \theta_e, \theta_d, \theta_{\mathbf{w}})$. $\theta_{\mathbf{w}}$ denotes the parameters of hypernetworks producing all the mixing weights and biases. $\lambda_g$ and $\lambda_{SD}$ are two scaling factors.

Although containing two mixing networks, the total mixing-net size of GoMARL is smaller than the commonly used single monotonic mixing network [27], as verified in Section 5.1. This is mainly attributed to the input dimension reduction. The monotonic mixing network takes the global state $s$ as input of all the hypernetworks. In contrast, we take specific group information (*i.e.*, individual group-related info $e$, group state $s^g$, and state $s$ is only used in one hypernetwork). This parameter reduction offsets the increase of an extra mixing network. Compared with flat value factorization methods learning only with the global state, our method allows fine-grained learning guided by more direct signals embedded in the policy gradients, facilitating intra- and inter-group cooperation.

# 5 Experiments

**Baselines.** We compare GoMARL with prominent baselines to verify its effectiveness and efficiency. Hu et al. [8] fairly compared existing MARL methods without code-level optimizations and reported that QMIX [27] and QPLEX [38] are the top two value factorization methods. The authors also finetuned QMIX (denoted as Ft-QMIX in our paper), which attains higher win rates than the vanilla QMIX. VAST [24] learns value factorization for sub-teams based on apriori setting on group number. Therefore, we compare GoMARL with Ft-QMIX, QPLEX, and VAST to show its performance as a value factorization method. The baselines also include role-based methods ROMA [39] and the representative credit assignment method RIIT [8]. The latter combines effective modules of noticeable methods, and comparing them can further illustrate the superiority of GoMARL.

**Experimental setup.** All the methods are trained with 8 parallel runners for 10M steps and are evaluated every 10K steps with 32 episodes. We report the 1st, median, and 3rd quartile win rates across 5 random seeds. Please refer to Appendix B for experimental setup details.

## 5.1 Performance on SMAC

Methods are evaluated on six challenging SMAC maps. The chosen maps involve homogeneous and heterogeneous teams with asymmetric battles, allowing a holistic study of all methods.

**Overall performance.** The performance comparison of all the methods in the six *Hard* and *Super Hard* SMAC maps is shown in Figure 5. As the results show, each baseline method only achieves satisfactory performance on some of the tasks with specific properties they specialize in; *e.g.*, RIIT performs well on MMM2 but converges much slower in other tasks; QPLEX's leaning is not efficient in 8m_vs_9m and corridor. Ft-QMIX significantly outperforms the vanilla QMIX and has more efficient learning than other baselines. GoMARL has a similar performance as Ft-QMIX in 8m_vs_9m and corridor. However, the superiority of GoMARL can be clearly validated in all the other maps.

**Parameter size for value mixing.** Many methods use larger mixing networks with stronger expressiveness and fitting abilities to obtain superior performance. However, they often fail when compared to baselines that use a mixing net of the same size [8]. As shown in Table 1, our dual-hierarchical mixing architecture has fewer parameters when there are a large number of agents (when $n > 5$). VAST is not compared since it utilizes a linear summation of local utilities. The results of GoMARL are the

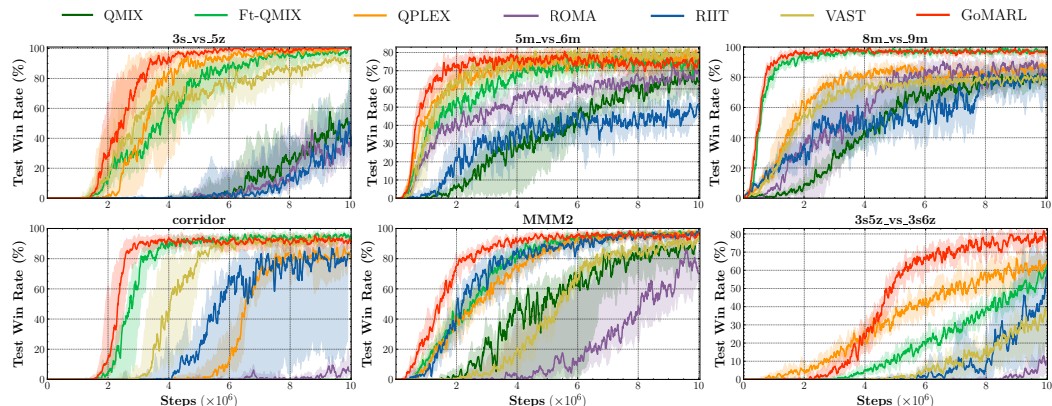

Figure 5: Comparison of GoMARL against baseline algorithms on six SMAC maps.

average of the five runs since each run may learn a slightly different grouping with various group numbers. GoMARL outperforms other methods despite using fewer mixing parameters, highlighting its inherent superiority over methods relying on stronger mixing networks.

Table 1: Size comparison of all methods' mixing network(s)

| Maps | (Ft-)QMIX | QPLEX | ROMA | RIIT | GoMARL |
|------|-----------|-------|------|------|--------|
| 3s_vs_5z | 21.601K | 72.482K | **13.281K** | 37.986K | 26.530K |
| 5m_vs_6m | 31.521K | 107.574K | **25.377K** | 51.362K | 31.554K |
| 8m_vs_9m | 53.313K | 197.460K | 63.393K | 93.986K | **51.427K** |
| corridor | 68.929K | 303.808K | 81.537K | 122.882K | **53.859K** |
| MMM2 | 84.929K | 342.248K | 134.401K | 177.282K | **74.244K** |
| 3s5z_vs_3s6z | 63.105K | 243.156K | 81.345K | 118.466K | **61.028K** |

**Component analysis and ablation study.** We next conduct detailed component studies to analyze how GoMARL improves efficiency and enhances performance. GoMARL contains three key components: (1) an automatic grouping mechanism that progressively divides the team into proper groups; (2) specialized agent networks that generate diversified policies while sharing all the parameters; and (3) sufficiently informative signals integrated into the gradients to promote efficient cooperation. We respectively study each module on three *Super Hard* maps to show how they influence performance.

(1.1) Ablation study of the grouping mechanism. We validate our grouping mechanism by comparing GoMARL with other intuitive alternatives. All methods utilize the same architecture as GoMARL to reflect how grouping itself influences performance. As shown in the top row of Figure 6, setting all agents as a group in corridor converges faster but has significant variances. It may be due to the hard exploration of efficient cooperation without grouping guidance. Another two intuitive groupings, each agent a group and an equal division into two groups $\{\{a_1, a_2, a_3\}, \{a_4, a_5, a_6\}\}$, have minor variance. However, their learning efficiency is affected since inappropriate groupings fail to promote cooperative behaviors. MMM2 contains heterogeneous agents; thus, a natural grouping is to keep the agents of the same type in one group. This natural grouping outperforms most alternatives but is inferior to our dynamic grouping adjustment. 3s5z_vs_3s6z is another scenario with heterogeneous agents, and the natural grouping of setting homogeneous agents as a group is also studied. It performs nearly the same as GoMARL because our mechanism learns the grouping precisely according to the agents' type in this scenario. We also analyze this map with a grouping containing three groups $\{\{a_1, a_4, a_5\}, \{a_2, a_6, a_7\}, \{a_3, a_8\}\}$ (1s2z, 1s2z, 1s1z); however, this balanced grouping fails to form effective collaboration. We can see from these results that appropriate grouping facilitates efficient cooperation and accelerates learning. Our grouping mechanism automatically learns adaptive grouping in different tasks and assists GoMARL with superior performance and efficiency.

(1.2) Learned grouping analysis. To demonstrate whether the learned grouping makes sense, we further visualize the trained strategy in a corridor battle, as illustrated in Figure 7. Six allied Zealots

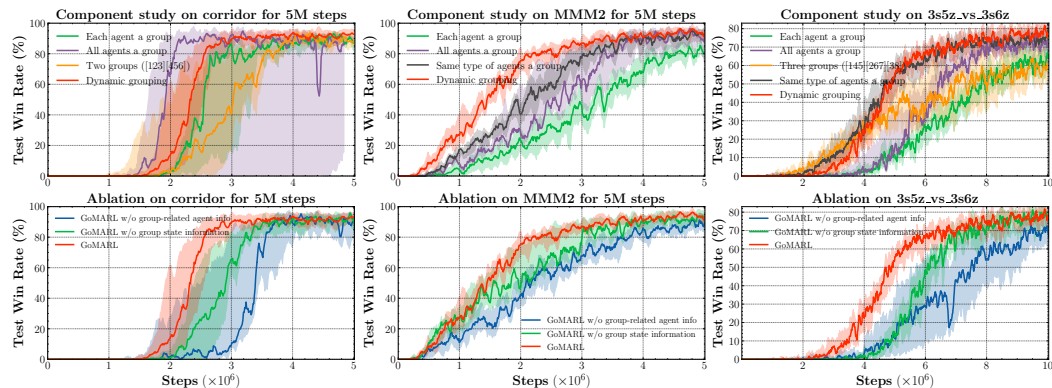

Figure 6: Ablations of the grouping mechanism (top row) and the group-related signals (bottom).

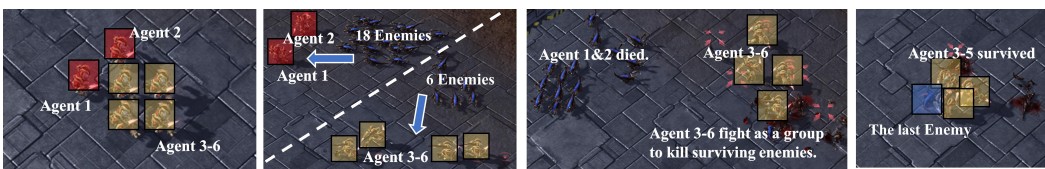

Figure 7: The learnt groups $\{\{a_1, a_2\}, \{a_3, a_4, a_5, a_6\}\}$ is explicable to the `corridor` combat situation.

fight twenty-four Zerglings on this *Super Hard* map. The massive disparity in unit numbers between the two sides implies that the whole team must refrain from launching an attack together. The only winning strategy is to sacrifice a small number of agents who leave the team and attract the attention of most enemies. Taking this opportunity, our large force eliminates the rest of the enemies. The surviving agents then use the same tactic to keep attracting several enemies and kill them together. In this battle, Agent 1 and 2 sacrifice themselves to attract most enemies and bring enough time for the team to eliminate the remaining enemies. Other agents fight as a subgroup and successfully kill all the surviving enemies. GoMARL learns a double-group setting in this battle, where Agent 1 and 2 are in the same group while the others are set in another group. This grouping is explicable in light of the combat situation, and this reasonable grouping guidance contributes to our superior performance.

(2) Component study of the specialized agents. We transplant our specialized agent network (SAN) into other baselines to verify module effectiveness. ROMA is not included since its actors are produced by its learned roles, and the replacement will invalidate the method. As shown in Figure 8, all methods are improved when equipped with SAN. However, even with SAN, all baselines fail to surpass GoMARL.

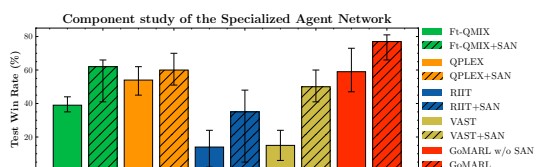

Figure 8: Method performance with and without SAN in `3s5z_vs_3s6z` after training 8M steps.

As shown in the learning curves in Appendix C, although dynamic grouping may reduce learning efficiency in the early stage of training, the learned grouping will significantly accelerate learning in the later stage. Therefore, the automatic grouping module and SAN are both crucial to GoMARL.

(3) Ablation study of the informative group-related signals. Our method models individual group-related information $e$ and group state $s^g$ to integrate them into the gradients. The former adjusts local utilities for cooperative behaviors and realizes policy diversity, while the latter fosters efficient group cooperation. We ablate them respectively to validate their effectiveness. Concretely, we replace the group state $s^g$ with the full state $s$ and remove the group-related information $e$ to degenerate the specialized agent networks into vanilla agent networks. As shown in the bottom row of Figure 6, both signals greatly improve learning efficiency. On the one hand, the agent info $e$ carrying group-related information is trained to encourage inter-group diversity, enabling extensive cooperative exploration and learning acceleration. On the other hand, the fused $s^g$ summarizes informative group-related status that facilitates efficient group-wise coordination. Although the global state $s$ provides complete

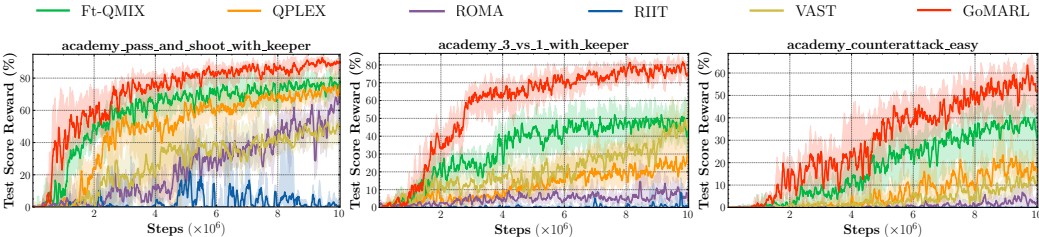

Figure 9: Performance comparison with baselines in three Google Research Football scenarios.

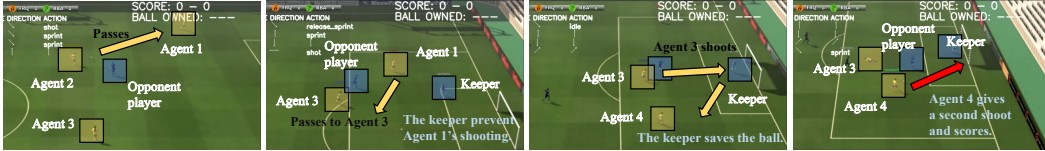

Figure 10: The learned groups $\{\{a_1, a_2\}, \{a_3, a_4\}\}$ is explicable to a `counterattack` situation.

knowledge, it is burdensome for agents to extract effective guidance that promotes policy learning. Therefore, utilizing our specific group-wise signals $e$ and $s^g$ is more efficient than the global state $s$.

## 5.2 Performance on Google Research Football

We also test GoMARL on three challenging Google Research Football (GRF) offensive scenarios. Agents in GRF need to coordinate timing and positions for organizing offense to seize fleeting opportunities, and only scoring leads to rewards. Therefore, the GRF tasks are more complicated than the SMAC battles, and the comparison in GRF is a secondary proof of our method's effectiveness.

**Performance.** Figure 9 shows the performance of all methods in GRF. GoMARL, Ft-QMIX, and QPLEX all perform well in `pass_and_shoot_with_keeper`. However, the advantages of our method become increasingly evident in the other two environments requiring more coordination. Only GoMARL achieves over 50% of the score reward in all scenarios. The superior performance with significant efficiency in the second testbed further demonstrates the transferability of our method.

**Visualization.** The trained strategy in `counterattack_easy` is visualized to validate if the learned group division makes sense. GoMARL learned a reasonable grouping $\mathcal{G} = \{\{a_1, a_2\}, \{a_3, a_4\}\}$ for this complex goal, in which the first group brought the ball into the penalty area through smooth coordination, while the second group created two shoots and the final goal through skillful cooperation, as shown in Figure 10. Appendix D provides a detailed discussion of this visualization.

## 6 Conclusion and Future Work

Grouping is essential to the efficient cooperation of multi-agent systems. Instead of utilizing apriori knowledge, this paper proposes an automatic grouping mechanism that gradually learns reasonable grouping as training proceeds. Based on the dynamic group division, we further model informative group-related signals to achieve fine-grained value factorization and encourage policy specialization, promoting efficient intra- and inter-group coordination. With these novelties, our method GoMARL achieves impressive performance with high efficiency on the SMAC and GRF benchmarks.

The automatic grouping mechanism of GoMARL is currently grounded in the value decomposition process, rendering it inapplicable to policy-based methods. Our future research will delve into automatic grouping based on policy gradients to enhance the learning efficiency of the policy-based methods. Furthermore, the current version of GoMARL is only designed for one-stage and one-level grouping, and it possesses the potential to achieve finer granularity in group division. We will investigate multi-stage and multi-level grouping (i.e., group-within-a-group) in the future to pursue better performance and higher learning efficiency in more complex environments.

# 7 Acknowledgement

This work is supported in part by the National Key Research and Development Program of China under Grant No. 2020AAA0103401; in part by the Natural Science Foundation of China under Grant 62076238, Grant 62222606, and Grant 61902402; and in part by the China Computer Federation (CCF)-Tencent Open Fund.

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

# A The Insight of the Group Shifting Scheme

The proposed automatic grouping mechanism dynamically adjusts the group division as the training proceeds. In the beginning, all agents belong to one group. As introduced in Section 4.1, we examine the learned $w_1^i$ of each agent every $c$ timesteps to check if the grouping needs adjustment. If there are agents who contribute little to their group, it indicates that these agents do not belong to their current group. All these selected agents are assigned to the following group until they appropriately contribute to where they belong. A new group is built for selected agents in the last group.

The proposed scheme ensures that agents who belong to the same group will not be misassigned to different groups after training. For the example in the right side of Figure 1, agent $a_i$ is first selected after $\alpha_1 \cdot c$ timesteps' training, and the initial grouping $\mathcal{G} = \{a_1, a_2, \cdots, a_n\}$ changes to $\mathcal{G}' = \{\{a_1, \cdots, a_{i-1}, a_{i+1}, \cdots, a_n\}, \{a_i\}\}$. Subsequently, after another training period, at timestep $\alpha_2 \cdot c$, two agents $a_j$ and $a_k$ in the first group are selected simultaneously to be moved out of the first group. At this point, they are automatically placed in the second group, *i.e.*, the group to which agent $a_i$ belongs, and $\mathcal{G}'' = \{\{\mathcal{A}_{-i,j,k}\}, \{a_i, a_j, a_k\}\}$. Instead of placing $a_j$ and $a_k$ in a brand new group, our group shifting scheme ensures that they have the opportunity to train with $a_i$ together in one group, determining if $a_i$, $a_j$, and $a_k$ (or two of them) are supposed to be in a group.

Later, agent $a_l$ in the first group and agent $a_i, a_k$ in the second group are chosen at timestep $\alpha_3 \cdot c$. The selection of $a_i$ and $a_k$ indicates that agent $a_j$ cannot cooperate well with them. Therefore, $a_i$ and $a_k$ are set in a new group (they should not return to the first group since they have been proved inappropriate for the first group), while $a_j$ stays alone in the second group. Agent $a_l$ from the first group is assigned automatically to the second group $\{a_j\}$. If $a_l$ is supposed to be with $a_i$ and $a_k$, it will be selected afterward. Because $a_j$ fails to form efficient cooperation with $\{a_i, a_k\}$, and if $a_j$ also cannot cooperate well with $a_l$, then $a_l$ will be selected later on and set in the third group $\{a_i, a_k\}$.

The group shifting scheme of GoMARL assigns the selected agents to their following group and ensures that each group is trained for a fixed period of cooperative attempts. As analyzed above, each agent can be grouped with appropriate agents that can efficiently work together after training.

# B Experiment Details

## B.1 Detailed Experimental Setup

We compare all the methods in six *Hard* and *Super Hard* StarCraft II micromanagement tasks (SMAC) [29] and three challenging Google Research Football (GRF) [16] scenarios. When resources are available, parallel training can facilitate faster convergence of methods [8] and is very common in RL and MARL communities [39, 45, 11]. GoMARL and all the baseline methods in our paper are trained with 8 parallel runners for 10M steps in both testbeds. Therefore, the timestep is not comparable to some papers that conduct experiments with only one "episode" runner. We use one NVIDIA Titan V GPU for training. We evaluate each method every 10K steps with 32 episodes and report the 1st, median, and 3rd quartile win rates across 5 random seeds. The detailed setting of GoMARL's hyperparameters is shown in our source code[1].

Many algorithms introduce implementation tricks when they are built. These code-level optimizations were studied in depth in [31, 48, 8] and were shown to have a significant impact on algorithm performance. To ensure fair comparisons, our experiments are based on the PyMARL2 [8] framework proposed to compare algorithms fairly. Please refer to PyMARL2's open-source implementation for further training details and fair comparison settings. In addition, some methods implement specific parameter tuning in diverse scenarios, and is unfair for comparison. Therefore, we fixed parameters of all methods for all scenarios in our experiments.

Lastly, adding vanilla-QMIX in our experiments is not intended to unfairly compare GoMARL and QMIX. After all, GoMARL is already better than the finetuned version of QMIX (denoted as Ft-QMIX), and there is no need to compare it with vanilla-QMIX to show our superiority. We added vanilla-QMIX's learning curves in Figure 5 only to avoid potential misunderstanding caused by the performance difference between Ft-QMIX in our paper and that of vanilla QMIX in other papers.

---

[1] `https://github.com/zyfsjycc/GoMARL`

## B.2 Detailed Information about SMAC Tasks

In each SMAC micromanagement problem, a group of units controlled by decentralized agents cooperates to defeat the enemy agent system controlled by handcrafted heuristics. Each agent's partial observation comprises the attributes (such as `health`, `location`, `unit_type`) of all units shown up in its view range. The global state information includes all agents' positions and `health`, and allied units' last actions and `cooldown`, which is only available to agents during centralized training. The agents' discrete action space consists of `attack[enemy_id]`, `move[direction]`, `stop`, and `no-op` for the dead agents only. A particular unit, Medivac, has no action `attack[enemy_id]` but has action `heal[enemy_id]`. Agents can only attack enemies within their shooting range. Proper micromanagement requires agents to maximize the damage to the enemies and take as little damage as possible in combat, so they need to cooperate with each other or even sacrifice themselves. We follow the default setup of SMAC in our experiments, and more settings, including rewards and observation information, can be acquired from the original paper [29] or open-source implementation.

Based on the performance of baseline algorithms, the tasks in SMAC are broadly grouped into three categories: *Easy*, *Hard*, and *Super Hard*. The key to winning some *Hard* or *Super Hard* battles is mastering specific micro techniques, such as *focusing fire*, *kiting*, avoiding *overkill*, et cetera. The battles can be symmetric or asymmetric, and the group of agents can be homogeneous or heterogeneous. Here we provide some characteristics of the scenarios to help gain insights into the good or poor performance of the methods:

- `3s_vs_5z` is a *Hard* asymmetric battle between two homogeneous teams. The allied Stalkers have to master the *kiting* technique and disperse in the area to kill the Zealots that chase them one after another. This map faces the delayed reward problem; however, it is not very strict about micro-cooperation between agents because of agents' scattering.

- The asymmetric scenarios `5m_vs_6m` and `8m_vs_9m` are two *Hard* maps offering a substantial challenge. The allied agents must learn to focus fire without overkill and to correctly position themselves with considerable precision to overcome the enemy team with more agents. The `5m_vs_6m` battle is relatively more difficult than `8m_vs_9m` due to fewer agents, so the enemy force with one more agent have a greater advantage.

- `corridor` is a *Super Hard* map that needs extensive exploration. Six allied Zealots fight twenty-four Zerglings on this task. The massive disparity in unit numbers between the two sides implies that the whole team cannot launch an attack together. The only winning strategy is to sacrifice a small number of agents who leave the team and attract the attention of most enemies, and the large force eliminates the rest of the enemies. The surviving agents then keep repeating the same tactic till the whole team kills all the enemies.

- `MMM2` is a representative *Super Hard* asymmetric battle between two heterogeneous teams with three kinds of units. One Medivac, two Marauders, and seven Marines have to battle against a team with one more Marine. Marauder has greater attack damage and health than Marine but with a longer `cooldown`. Medivac has no damage but can heal other agents.

- `3s5z_vs_3s6z` is a *Super Hard* map that requires breaking the bottleneck of exploration, where three Stalkers and five Zealots battle against three Stalkers and six Zealots. This map requires more cooperation between agents and is the most challenging task in SMAC.

## B.3 Detailed Information about GRF Academy Scenarios

Google Research Football (GRF) Academy includes several scenarios which can be commonly found in football games. Agents need to coordinate timing and positions for organizing offense to seize fleeting opportunities, and only scoring leads to rewards. Each agent's partial observation contains the absolute positions and moving direction of the ego-agent, relative positions and moving directions of other agents, and the ball. The global state information includes the absolute positions and directions of all agents and the ball. Agents have a discrete action space of 19, including moving in eight directions, three kinds of ball pass, two kinds of dribble, two kinds of sprint, shot, sliding, stop-moving and do-nothing. Proper cooperation requires agents to pass and shoot effectively with a good direction at proper timing, thus making GRF scenarios much more complicated than SMAC's. Other details can be acquired from the original paper [16] or the open-source implementation.

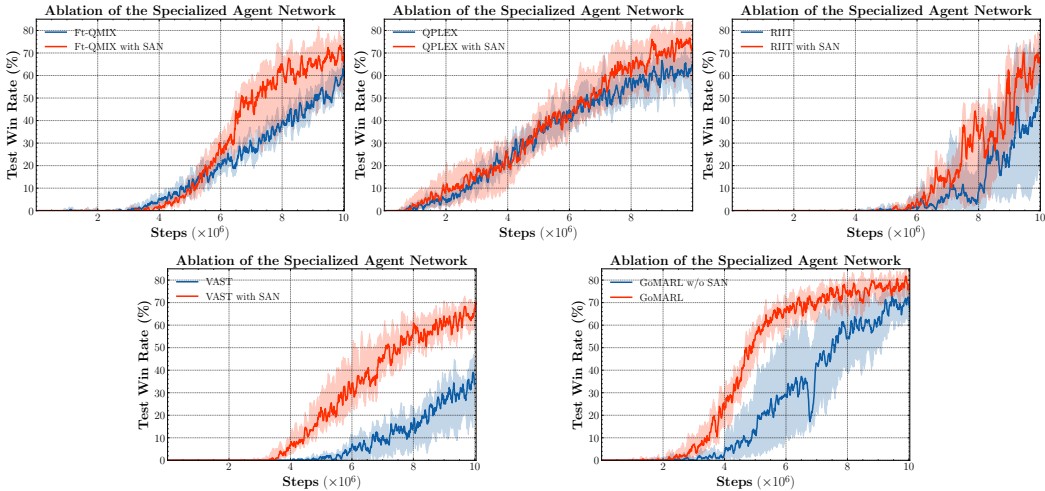

Figure A11: Method improvement with the proposed SAN in the *Super Hard* map `3s5z_vs_3s6z`.

Here we provide an introduction of the three scenarios we utilized to help gain insights into the good or poor performance of all the methods in our paper:

- `academy_pass_and_shoot_with_keeper` includes two of our players trying to score from the edge of the box. One of our agent is on the side with the ball and next to a defender. The other agent is at the center, unmarked, and facing the opponent keeper.
- `academy_3_vs_1_with_keeper` contains three of our agents and two opponent players (a defender and a keeper). Our agents try to score from the edge of the penalty area. One stands in the middle, while the others are located on both sides of the area. Initially, the agent at the center keeps the ball and directly faces the defender.
- `academy_counterattack_easy` contains four of our agents and two opponent players (a defender and a keeper). Agents are initialized far from the penalty area and stand evenly in an arc centered on the goal. The second agent from the top initially keeps the ball and has to pass it to a teammate at the appropriate time to avoid interception.

## C   Performance of Methods with and without the Specialized Agent Network

The proposed specialized agent network (SAN) is highly transferable. To further validate its effectiveness, we perform the specialized agent network on other baseline methods in Section 5.1 to see if they can perform better. ROMA is not included in this study since ROMA's agent networks are produced by its learned roles, and the replacement will invalidate the main idea of ROMA. As shown in Figure 8, our specialized agent networks significantly improve methods' performance and learning efficiency in the *Super Hard* SMAC task `3s5z_vs_3s6z`. Compared to agent networks utilizing the vanilla parameter-sharing mechanism that limits policy diversity, our specialized agent network adjusts the local utility with the individual group-related information and enables various styles of behaviors related to the group status, encouraging extensive exploration and accelerating learning. The complete learning curves are shown in Figure A11.

As Figure A11 illustrates, the performance of all methods is further improved when equipped with our specialized agent network. Specifically, the learning efficiency of Ft-QMIX is boosted. The variance of RIIT is markedly reduced, and the win rate is increased by about 10%. The improvement of VAST is the most obvious; both learning efficiency and win rate are enhanced, and the variance is very clearly reduced. QPLEX's improvement is not very obvious; however, it obtains a slightly higher learning speed and achieves the highest win rate among all the baseline methods.

Most importantly, even equipped with our specialized agent network, all baseline algorithms fail to surpass GoMARL in terms of learning efficiency and the final win rate. GoMARL with vanilla agent networks looks much inferior to GoMARL with the specialized agent network, so it is worth questioning whether the dynamic grouping module is ineffective. The dynamic group learning may

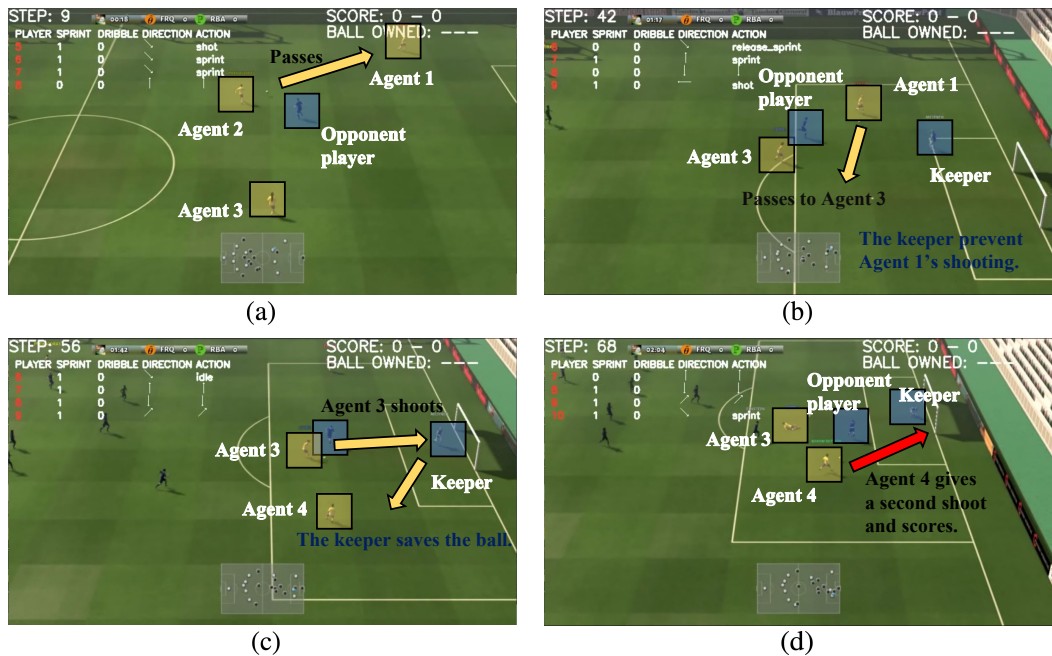

Figure A12: A visualization of learned policies of `academy_counterattack_easy`. The yellow arrows show the motion of the ball, and the red arrow illustrates the scoring shoot.

reduce efficiency to a certain extent in the early stage of training compared to QPLEX, however, in the middle and late training stages, the learned grouping will have a significant effect when equipped with the proposed specialized agent network. Therefore, the two modules of GoMARL, the automatic grouping module and the specialized agent network, are both crucial to the performance of GoMARL.

## D  Visualizations of the Learned Policies in GRF

The trained strategy in Google Research Football is visualized to validate if the learned grouping makes sense. In Section 5.2, we visualize a match of `academy_counterattack_easy` with a more complex strategy to prove the rationality of the grouping GoMARL learned. Here, we provide a detailed discussion of the visualization in Figure A12 (a zoom-in version of Figure 10 for clarity).

As shown in Figure A12(a), Agent 2 holds the ball at the beginning and faces an opponent rushing toward him. Agent 2 passes the ball to Agent 1 to prevent the ball from being stolen. Subsequently, in (b), Agent 1 carries the ball and tries to break through. However, the opponent goalkeeper blocks his attacking route, and Agent 1 continues passing after a short carry. Agent 3 makes a good run and catches the ball smoothly but shoots quickly in Figure A12(c) since the opponent is close. The goalkeeper easily saves this hasty attack. Agent 4 in (d), who learns excellent coordination with Agent 3, stops the ball and immediately adds another shot to create the goal.

GoMARL's automatic grouping module learns a reasonable grouping $\mathcal{G} = \{\{a_1, a_2\}, \{a_3, a_4\}\}$ for this complex goal, in which the first group successfully brings the ball into the penalty area through smooth coordination, while the second group creates the final goal through skillful cooperation.

