# OpenReview forum: "Automatic Grouping for Efficient Cooperative Multi-Agent Reinforcement Learning"
_NeurIPS.cc/2023/Conference — NeurIPS 2023 poster_

### Official Review · Reviewer_Dgdo · 2023-06-12

**Soundness:** 3 good
**Presentation:** 2 fair
**Contribution:** 3 good
**Rating:** 7
**Confidence:** 3

**Summary:**

This paper claims that grouping is beneficial for the efficiency of team coordination. To verify this claim, the authors propose a new learning framework called GoMARL, which achieves efficient cooperation by learning automatic grouping without domain knowledge. The joint action-values are factorized in a group-oriented way, which makes this work different from other related works. The group is generated dynamically and automatically. Meanwhile, a policy learning strategy is proposed to utilize the proposed grouping mechanism. The aim of this policy is to let agents in a same group specialize in similar policies, and agents in different groups have diverse strategies. The proposed solution is evaluated in StarCraft II micromanagement tasks and Google Research Football.

**Strengths:**

The overall idea of introducing grouping into multi-agent value function factorization is quite novel. I’m also happy to see that it is executed well in many challenging scenarios.

The authors also provide detailed case studies of the learned groups and their performance in both SMAC and Google Research Football. I find these materials very helpful to show that grouping is helpful in multi-agent cooperation tasks.

The experiments are also very detailed, showing that GoMARL performs well on multiple domains. I highly appreciate the authors’ efforts in designing ablation studies on each component of their proposed solution, which makes it very intuitive to verify the contribution of each component.


**Weaknesses:**

Although the overall idea of grouping is intuitive, it is quite challenging to grasp the detailed implementation procedures. For instance, in Line 139: “if agent $a_{j_{i}}$ in group $g_j$ establishes consistency with all the other agents $a_{-j_{i}}$ in group $g_j$, …, then $a_{j_i}$ should not be in group $g_j$ anymore”, this statement represents the standard of grouping, but it is quite abstract. Is it supported by natural systems on grouping or is it a compromise due to the restriction of GMM principle?

Some statements can be rephrased in a more straightforward manner. For instance, in Line 270: “GoMARL maintains two mixing networks, and if it has more parameters for value mixing is a natural question”, this representation is quite ambiguous.


**Questions:**

Grouping is common in reality, but it also restricts the whole policy space into a sub-space where the requirements of grouping holds. Will this restriction influence the overall outcome in some cases?

**Limitations:**

The authors do not appear to discuss the limitations of their work. I suggest that the authors discuss more scenarios where grouping is beneficial and where the contribution of grouping is limited.

---

> ### Author Rebuttal · Authors · 2023-08-09
>
> We sincerely thank the reviewer for the insightful review and helpful comments. Following are our responses to all your questions.
>
> **Q1: Ambiguity in Line 139.**
>
> A1: We are sorry for any confusion caused. The automatic grouping mechanism we propose in this paper is guided by the GGM principle, rather than being derived from natural systems. This insight motivates our "select-and-kick-out" grouping mechanism design. The detailed implementation procedures are illustrated in Lines 142-147 and the clear example in Fig.1(right). We highly recommend the reviewer refresh these parts, as they provide comprehensive insights into the implementation details. Appendix B (the insight of the group shifting scheme) is also beneficial to understand the grouping mechanism details. We have revised the statement in Line 139 to eliminate the ambiguity. Thank you very much for pointing out this unclear expression.
>
> **Q2: A statement should be rephrased in a more straightforward manner.**
>
> A2: We sincerely appreciate this helpful comment. We revised the statement you pointed out in Line 270: "GoMARL maintains two mixing networks but employs fewer mixing parameters compared to the baseline methods, particularly in scenarios with a large number of agents. GoMARL outperforms other methods despite using fewer mixing parameters, highlighting its inherent superiority over methods relying on stronger networks". Additionally, we thoroughly checked our paper and rephrased essential statements in a clear and straightforward manner to eliminate any ambiguity.
>
> **Q3: Whether GoMARL restricts the whole policy space into a sub-space?**
>
> A3: While GoMARL involves group division in the multi-agent system, all agents, regardless of their group, still explore and learn within the entire policy space.
>
> We want to emphasize that GoMARL is a value decomposition method under the CTDE framework. We learn group division only as a reasonable way to utilize the group-wise information to estimate fine-grained value functions and facilitate cooperation among agents. Hence, there are neither restrictions on the overall policy space nor requirements that agents can only explore in the group-wise policy spaces.
>
> The automatic grouping mechanism based on the GGM principle also has no restriction on the policy space. As for the joint action space of the multi-agent system, each agent's action selection is driven and learned by $Q_{tot}$ in GoMARL. As for the agent's individual action space, no limitations related to the agent's group are imposed; agents continue to learn and make decisions within their complete action spaces.
>
> Therefore, GoMARL's grouping mechanism does not restrict the entire policy space into sub-spaces and does not influence the overall outcome.
>
> **Q4: Discussion on limitations.**
>
> A4: Thank you for suggesting including a separate subsection of limitations. It is very constructive to further improve the quality of our paper.
>
> Here we present our discussion on limitations and future plans: "The automatic grouping mechanism of GoMARL is currently grounded in the value decomposition process, rendering it inapplicable to policy-based methods. Our future research will delve into automatic grouping based on policy gradients to enhance the learning efficiency of policy-based methods. Furthermore, the current version of GoMARL is only designed for single-level grouping, and it possesses the potential to achieve *finer granularity in group division*. We will investigate multi-level grouping (*i.e.,* group-within-a-group) in the future to achieve nested group division, pursuing better performance and higher learning efficiency in more complex environments."
>
> **Thank you once again for your insightful review and constructive comments. We hope our response can address all your concerns. It would be greatly appreciated if you would re-evaluate our paper based on our responses. If there are any remaining uncertainties, we welcome new comments to ensure clarity and satisfaction during the discussion phase.**

---

> > ### Comment · Reviewer_Dgdo · 2023-08-18
> >
> > Thank you for your response! I have updated my score accordingly.

---

> > > ### Author Response · Authors · 2023-08-19
> > > **Thank You**
> > >
> > > We sincerely thank you for reviewing our response and updating the score. We are delighted to see that our reply has effectively addressed all of your concerns. We thank you once again for your constructive feedback and contributions to our paper.

---

### Official Review · Reviewer_wS3D · 2023-06-14

**Soundness:** 2 fair
**Presentation:** 3 good
**Contribution:** 3 good
**Rating:** 6
**Confidence:** 4

**Summary:**

This paper proposes an automatic grouping mechanism and a GGM principle to build a value function factorization. And the authors model informative group-related signals to encourage specialization in policies and inter-group coordination. The authors evaluate the performance of all methods on the SMAC and GRF benchmarks.

**Strengths:**

1. The idea of grouping agents is sound.
2. The paper is well-written and easy to follow.

**Weaknesses:**

1. As the authors claim that the method is capable of the correct selection of non-optimal (sacrificial) actions, does this imply that the value decomposition enables the representation of various types of payoff matrices and facilitates decentralized execution? If this is the case, it would be worthwhile to discuss how this method can potentially address the issue of relative overgeneralization. Additionally, it might be beneficial for the authors to test their approach on a simple matrix game, such as

   |      | a_1 | a_2 |a_3|
   |---------|---------|---------|---------|
   | a_1 | 3 | -7 |   -5|
   | a_2 | -7 | 0 |   0|
   | a_3 | -5 | 0 |   2|

2. The absence of a theoretical analysis regarding the representation capacity of the proposed value decomposition weakens the contribution of the paper. It would greatly enhance the strength of the work if the authors could provide a thorough theoretical analysis to support and validate the effectiveness of their value decomposition approach.
3. What is the difference between e^i and role in other role-based methods?
4. The performance of ROMA and RODE is very poor. Could the author give a further explanation why role-based methods are not suitable for these tasks?


**Questions:**

Please answer the questions in the weaknesses.

---

> ### Author Rebuttal · Authors · 2023-08-09
>
> We sincerely appreciate the insightful comments and your efforts in reviewing our paper. Following are our responses to all your concerns.
>
> **Q1: Value Decomposition Representation.**
>
> A1: GoMARL is not a method that aims to resolve the representation constraints on value mixing. Instead, it is a value factorization framework designed to facilitate efficient cooperation by learning group divisions.
>
> The term "sacrifice" does not imply that we enhance the expressiveness of the mixing network. Before convergence, the still-learning $Q_i$ might be unable to guide agents to take actions that maximize team benefits. Hence, agents need to "sacrifice" their *current benefits*, and opt for actions with smaller *present $Q_i$*, to explore greater team gains and learn the optimal action. We hope the above statement can clarify any misunderstanding regarding the term "sacrifice."
>
> Although GoMARL does not focus on expressiveness, it is a general group-wise framework for value factorization. Existing methods to overcome representation constraints can be easily integrated with GoMARL. Notably, GoMARL will not influence the inherent expression capacity of the base mixing scheme. By adopting QPLEX's advantage mixing, GoMARL effectively learns optimal actions in the provided matrix:
>
>    |  | $u_1$    | $u_2$ | $u_3$ |
>    | ----- | -------- | ----- | ----- |
>    | $u_1$ | **3.03** | -6.29 | -4.38 |
>    | $u_2$ | -6.49    | 0.07  | 0.08  |
>    | $u_3$ | -4.00    | 0.11  | 2.12  |
>
> This also clarifies the absence of a theoretical analysis regarding representation capacity. GoMARL, like RODE and ROMA, does not focus on theoretical analyses of representation. Instead, GoMARL is oriented towards further studying the relationship between individuals and the team. Extensive component studies show how grouping works in GoMARL, which are also very convincing (highly recognized by all the other reviewers).
>
> **Q2: Comparison between $e_i$ and role.**
>
> A2: Thank you for this insightful question. We hope this response can assist you in better grasping GoMARL's motivation and distinct advantages. We first emphasize that $e_i$ does not represent the learned group; instead, it is the group-related information that summarizes the agent's *local* history *from the group perspective* (Line 188).
>
> * Compared with role in RODE
>
> The role in RODE is essentially a cluster of action representation, with the **number of roles being predefined**. RODE learns action representations through **supervised learning** and clusters them into $k$ (a hyperparameter) roles. The role representations are derived by averaging the action representations within each role. RODE determines each agent's role by selecting the role representation that maximizes the local $Q_i$ value.
>
> GoMARL and RODE are both thoughtfully designed methods. However, it's important to note that $e_i$ diverges from the role in RODE. $e_i$ is motivated by novel insights, providing informative knowledge for fine-grained value decomposition. Unlike RODE, $e_i$ does **not rely on apriori knowledge** and is trained in an **end-to-end** manner.
>
> * Compared with role in ROMA
>
> Common Ground: The only similarity between $e_i$ and role $\rho$ in ROMA is their generation through an encoder-decoder architecture—a common practice for information encoding. Beyond this architectural similarity, $e_i$ and $\rho$ are entirely different.
>
> Differences: The grouping $G$ (not $e_i$) of GoMARL is driven by $Q_{tot}$, and the group division is achieved during the value decomposition process. $e_i$ is the group-wise information integrated into policy gradients to promote intra-group coordination. On the other hand, $\rho$ in ROMA is an embedding of the encoded local observation. Unlike $e_i$, it is not aimed at promoting intra-role coordination. In addition, different from the specialized regularizer in ROMA, our SD-regularizer is set to extract  *group-related* information from the *individual* history. In summary, $e_i$ and $\rho$ have entirely distinct motivations, insights, and purposes.
>
> **Q3: Performance of RODE and ROMA.**
>
> A3: **RODE**'s role selection highly relies on the environment. Factors such as the predefined role numbers $k$, supervised learning iterations, and the way to generate role representation, are acutely attuned to the particular environment. This environmental sensitivity may render agents incapable of executing crucial actions, owing to unreasonable action space divisions. That is why RODE performs well in SMAC but fails in GRF. In addition, RODE uses several rules based on prior knowledge[1], which leads to an unfair comparison. Thus under our fair comparison (detailed in Appendix C.1), RODE's performance may sometimes be unstable.
>
> **ROMA** learns each agent a role, with all regularizers acting directly on the role. It *implicitly* guides policy learning through three attributes (dynamic, identifiable, and specialized). However, ROMA lacks explicit mechanisms that directly stimulate cooperative behaviors. In contrast, GoMARL learns multiple modules based on the groups, explicitly facilitating efficient intra/inter-group cooperation. As a result, GoMARL is more efficient than ROMA.
>
> The performances of RODE and ROMA in our paper are one of the best among public papers, better than their performances in papers from the same research team [2,3] and other teams [1,4,5,6], with no unfair comparisons.
>
> [1] LDSA: Learning dynamic subtask assignment in cooperative multi-agent reinforcement learning. NeurIPS, 2022.
>
> [2-6] are NeurIPS and ICML papers listed in Appendix Line 95. (omitted here due to rebuttal length restriction)
>
> **We hope our response can address your concerns, clarifying our main contribution (which does not center on solving the representation problem). Your *re-evaluation* of our paper is critical to us, and we genuinely appreciate your consideration. If you have any new concerns, we would be glad to further discuss them to ensure clarity in the next phase.**

---

> > ### Comment · Reviewer_wS3D · 2023-08-15
> >
> > Thank you for the response. Now, I understand the contribution more clearly, and I will raise the score accordingly.

---

> > > ### Author Response · Authors · 2023-08-15
> > > **Thank you**
> > >
> > > Thank you very much for reviewing our response and updating the score. Once again, we sincerely appreciate your thorough review and insightful feedback.

---

### Official Review · Reviewer_y5k7 · 2023-06-20

**Soundness:** 3 good
**Presentation:** 3 good
**Contribution:** 4 excellent
**Rating:** 7
**Confidence:** 4

**Summary:**

This paper considers fully cooperative MARL in the CTDE setting. Specifically, this work focuses on the notion of groups and proposes three components to improve learning performance:
- specialized agent network (SAN): relies on parameter-sharing across agents, but the shared policy is group conditioned and regularized such that it is similar for agents in a same group and different across groups.
- three stage value mixing: (1) w_1, from agent-level to group-level with agent information. (2) w_2, from group-level to group-level, but incorporating group information. (3) from group-level to total-level (mixing weights and the information used is unclear).
- automatic and dynamic group formation during learning.

The resulting method GoMARL is compared against several baselines on two environments and several challenging tasks. It is shown to outperform the baselines. Authors also propose ablation to investigate the importance of SAN, incorporating different level of information in the value mixing, and automatic group formation. Finally, the learned groups and strategies are qualitatively analyzed and discussed by looking at how agents behave in selected scenarios.


**Strengths:**

- Improving performance of cooperative MARL in the CTDE setting is of significant importance for the research community.
- Relying on group for cooperation and coordination is interesting and well motivated.
- The paper is well written and does a good job at explaining the method despite its complexity.
- Empirical evaluation is thorough and highlight significant and reliable performance improvements.
- The method relies on less network parameters than the baselines.
- SAN improvement is transferable to other methods.

**Weaknesses:**

- Clarity: I have trouble understanding several points related to the GGM principle.
    - l.176: I cannot understand how "sacrificing self-interest for greater team benefit" could occur in the fully cooperative setting (shared reward) since individual agent's reward and team reward are exactly the same. If an agent killing itself increases the team reward, it would thus be incentivized to do so by its own individual reward. Therefore, I do not understand why consistency would be lost inside the group:
        - Definition 2: I do not understand why individual-group consistency is lost.
        - l.140: I do not understand why one would enforce group inconsistency by kicking out consistent agents out of the group.

- Limitation: I think that the assumption that agents in a group must have similar policies while being different to agents in another group could be overly restrictive. Imagine the setting of two groups each building half of a bridge: both groups need varied roles (resource gatherer, architect and builder for instance). My understanding is that grouping in that case is still relevant for credit assignment but impedes policy specialization inside each group.

- Clarity: Section 4.3. The method is quite complex and has lots of moving part, so I believe it would be very valuable to summarize it by explicitly expressing Q_tot in terms of individual Q and highlighting the different mixing levels, weights and the information used (agent's, group's, total, etc.). For instance, on what are the total mixing weights conditioned?

- Comparison: I think it is important to report GoMARL performance against MAPPO's given that [1] considers the same environments. Also, MAPPO's centralized value function could leverage GoMARL to see if it improves credit assignment and performance.

**Questions:**

- Figure 1.: shouldn't G''' be {{A}, {a_i, a_l}, {a_j, a_k}} since a_i wasn't selected but a_j was?
- I found using $a$ as agent index a bit confusing, as usually $a$ is for action and agent index is $i$.
- could it be that no stable grouping is achieved? Like varying team strategy or any other reason?
- equation (4): does the h correspond to the same sample, i.e. the same situation but observed by different agents, or does it also correspond to different samples of the replay buffer?

### Summary
I think this is a very good paper, yet I would like to have a better understanding of it before increasing my score. Therefore, I encourage authors to address my concerns (see. Weaknesses).

### references
[1] Yu, Chao, et al. "The surprising effectiveness of ppo in cooperative multi-agent games." Advances in Neural Information Processing Systems 35 (2022): 24611-24624.


### EDIT
I have updated my score from 6 to 7 post-rebuttal

---

> ### Author Rebuttal · Authors · 2023-08-09
>
> Thank you very much for your thorough reading and insightful comments. Following are our responses to all your concerns.
>
> **Q1: Ambiguity of the GGM principle.**
>
> A1: We apologize for the unclarity. We agree with your view on consistency under fully cooperative settings, and we also follow the consistency principle when implementing GoMARL (Eq.3). However, in the initial training phase or before convergence, the still-learning $Q_i$ may not yet be able to guide the agent to take a series of actions to maximize team benefits. As in your example, the action that maximizes the eventual $Q_i$ is "suicide", but it may not be before training ends; Hence, the agent needs to "sacrifice" its *current benefits during training*, and opts for actions with smaller *present $Q_i$*, to explore greater team gains (to learn suicide). Our viewpoint is to integrate group-wise information into the gradient (Eq.3) to help learn the optimal action (suicide in your example) and enhance collaboration within groups. On the other hand, if an agent $a_i$ perceives no necessity to "sacrifice", i.e., $Q_i$ has consistency with $Q_{group}$, then $a_i$ can leave this group. We are sorry for not clearly expressing our thoughts. We have revised our expression in this part and hope our response can solve your doubts.
>
> **Q2: Example of building two separate half bridges.**
>
> A2: GoMARL can handle this example. In specific, by pursuing policy similarity within groups and policy diversity between groups, we can attain a finer-grained grouping of 6 groups (the resource gatherer group $g_{rg}^1$, architect group $g_{a}^1$, and builder group $g_{b}^1$ in half-bridge 1; resource gatherer group $g_{rg}^2$, architect group $g_{a}^2$, and builder group $g_{b}^2$ in half-bridge 2). Even if there is only one architect, GoMARL can still learn a separate group for the architect. With the above reasonable grouping, the intra/inter-group coordination will significantly improve the overall team efficiency. Therefore, our SD-regularizer showcases relatively broad applicability.
>
> **Q3: Details of mixing levels, weights, and the information used.**
>
> A3: Thank you for this valuable suggestion. We hope the detailed explanation of the mixing weights and the information utilized can facilitate a better understanding of GoMARL.
>
> (1) First mixing level:
>
> $w_1^i=f^i_{w_1}(h^i;\theta^i_{w_1})\in\mathbb{R}^{1\times k}$, where $k$ is the embedding size; and the group-wise $W_1^g\in\mathbb{R}^{n_g\times k}$ is the stack of $w_1^i$ for all $n_g$ agents $a_i$ in group $g$.
>
> $w_2^g=f^g_{w_2}(s^g;\theta^g_{w_2})\in\mathbb{R}^{k\times 1}$, where $s_g$ is the group-wise state of group $g$ (details in Lines 223-227).
>
> Insights of $w_1$ and $w_2$: The automatic grouping relies on $w_1$. Stacking individual $w_1$ of agents within the same group ensures the dynamic changes in group size (agent number per group). $w_2$ incorporates group-wise information. GoMARL generates $Q^g_{group}$ by mixing $Q^i$ through $w_1$ and $w_2$.
>
> (2) Second mixing level:
>
> $w_3=f_{w_3}(s^g;\theta_{w_3})\in \mathbb{R}^{1\times k}$; and $W_3 \in \mathbb{R}^{m \times k}$ is the stack of $w_3$ for all $m$ groups.
>
> $w_4=f_{w_4}(s;\theta_{w_4})\in \mathbb{R}^{k\times 1}$.
>
> Insights of $w_3$ and $w_4$: The second mixing level has a similar architecture as the first level. $w_3$ conditions on the group-wise information and enables dynamic changes in the group number, while $w_4$ incorporates global state information $s$ and generates $Q^{tot}$.
>
> Although containing two mixing levels, GoMARL relies on fewer mixing parameters than the baselines (Appendix D).
>
> **Q4: Comparison with MAPPO.**
>
> The requested comparison is shown in **the PDF of the global Rebuttal Comment**. MAPPO was not included in our initial paper because it is not a value factorization method but a policy-based method. Notably, MAPPO is published in Track on Datasets and Benchmarks. It is a valuable *engineering* work that thoroughly studies five implementation factors (tricks) of PPO in the multi-agent setting. Among them, adjusting "input representation to value function" involves *changes to the original SMAC environment*, which causes *unfair comparison* when compared with other methods. Thus, we compare GoMARL with MAPPO-Environment-Provided (i.e., MAPPO-EP in the original paper). GoMARL (with no trick) outperforms MAPPO (with four tricks) in all six SMAC maps.
>
> GoMARL is a value decomposition method where agents select their actions based on $Q_i$; however, the policy-based method MAPPO does not use $Q_i$ for the value estimation. GoMARL is currently grounded in the value decomposition process, rendering it unhelpful to policy-based methods. Following your valuable suggestion, our future research will delve into automatic grouping based on policy gradients to enhance the learning efficiency of policy-based methods.
>
> **Q5: Response to the "Questions" list.**
>
> A5: Following are responses to the "Questions" list. Due to rebuttal length limitations, we do not repeat your questions here.
>
> 1. Highlights with purple in Fig.1 denote the agents that *have adjusted groups*. $a_l, a_i, a_k$ adjusted their groups at the third iteration, and $G'''=\\{ \\{ A \\},\\{ a_j, a_l \\},\\{ a_i, a_k \\}\\}$ is the new grouping after adjustment. Please see the detailed explanation in Appendix B.
>
> 2. We will follow your advice to change the agent index from $a$ to $i$.
>
> 3. GoMARL gradually learns a stable grouping during training, with no instability observed in experiments; Each task can be solved with multiple strategies, and GoMARL may learn different groupings based on the map context.
>
> 4. $h$ in Eqn.4 corresponds to the same situation observed by different agents.
>
> **Once again, we appreciate your thorough review and helpful comments. If we address your concerns, we kindly ask for your consideration in increasing the score, as you mentioned. Should any questions remain unresolved, we welcome further discussions in the next phase.**

---

> > ### Comment · Reviewer_y5k7 · 2023-08-14
> > **Thank you**
> >
> > I thank the authors for their response and I have updated my score accordingly.

---

> > > ### Author Response · Authors · 2023-08-14
> > > **Thank you**
> > >
> > > Thank you very much for reviewing our response and updating the score. Once again, we genuinely appreciate your meticulous reading and insightful comments.

---

### Official Review · Reviewer_p19m · 2023-07-07

**Soundness:** 3 good
**Presentation:** 4 excellent
**Contribution:** 3 good
**Rating:** 7
**Confidence:** 3

**Summary:**

This work introduces group-oriented MARL (GoMARL). GoMARL leverages a grouping mechanism that automatically creates sub-teams in a cooperative multi-agent team in an end-to-end fashion by maximizing the expected global return.

In my opinion this approach is a nice, balanced solution for CTDE frameworks where the issues of fully centralized and decentralized models are greatly alleviated by instead looking at the coordination problem at a sub-team level, rather than individual agents, or one giant connected team. A major contribution and perhaps the most interesting part of the approach is the automatic grouping which does not require domain knowledge or a priori determined group number. The paper is very well-written and is easy to follow and understand. Extensive evaluations and ablation studies are carried out which confirms the applicability and efficiency of the approach.

I only have a few minor comments and questions:

- Since automatic grouping is being tackled, can this approach be directly applied to cooperative heterogeneous teams where each agent has a class-specific action-space (or observation-space) and objective. In such scenarios, despite having local separate objectives, classes of agents must collaborate towards a greater global objective/mission. Can this approach be reliable enough to include such class-dependent grouping integrated, for instance when specific classes/types of agents must be grouped for the whole thing to work?

- The process of end-to-end learning for automatically clustering agents can be highly complex and computationally expensive. I can see that with increasing number of agents, the possible combinations of sub-groups could grow exponentially, making the task of finding the optimal group indexes a significant challenge. Can authors comment on this please?

- Can inter-agent communication be leveraged here, for instance as a part of input/observation when needed? Would it require a similar locality and grouping?

- Limitations of the proposed approach are never discussed in the paper. Please add a separate subsection at the end to cover potential known limitations of the method.


**Strengths:**

See above

**Weaknesses:**

See above

**Questions:**

See above

**Limitations:**

No. Limitations are not discussed.

---

> ### Author Rebuttal · Authors · 2023-08-09
>
> Thank you very much for your constructive feedback and acknowledgment of our efforts. Following are our responses to all your concerns.
>
> **Q1: Applicability of automatic grouping to heterogeneous teams.**
>
> A1: GoMARL can be directly applied to cooperative heterogeneous teams. This paper focuses on fully cooperative tasks as Dec-POMDP (Lines 92-99), and this setting has no local separate objectives. Classes (groups) of agents trained by GoMARL collaborate towards the global objective (global reward for the team) since our grouping mechanism and the individual/group Q-values are learned by maximizing the expected global return.
>
> The two super hard scenarios showcased in Fig.4, 3s5z_vs_3s6z and MMM2 (involving agents with distinct action spaces), are heterogeneous settings where GoMARL outperforms baseline methods significantly. The efficacy of our grouping mechanism lies in its ability to adapt to varying task contexts. For example, agents are grouped according to their class/type in 3s5z_vs_3s6z (i.e., $G = \\{\\{3s\\},\\{5z\\}\\}$), while in MMM2, GoMARL clusters agents based on task context (Fig.5 and Lines 297-306). Notably, GoMARL automatically learns the grouping, but it is also reliable when facing predefined grouping requirements (like your request that specific classes/types of agents must be grouped). This can be achieved by initializing agents that must be grouped within the same subgroup and fixing this subgroup throughout training.
>
> **Q2: Increasing agent numbers results in exponentially growing sub-group combinations, posing a challenge in finding optimal group indexes.**
>
> A2: Our automatic grouping mechanism diverges from conventional search algorithms. Rather than seeking optimal indexes across all sub-group combinations, GoMARL employs the "*select and kick-out*" scheme (as illustrated in Lines 146-147 and Fig.1.right) based on the proposed GGM principle. With each new agent $a_i$, GoMARL determines whether it belongs to the current group $g_j$ at every timestep. This extra assessment is essentially an additional evaluation of $w_1^i$. Therefore, GoMARL's computational cost scales linearly with the number of agents, ensuring significant learning efficiency.
>
> **Q3: Integrate inter-agent communication into GoMARL.**
>
> A3: This paper focuses on cooperative Dec-POMDP without communication mechanisms. However, GoMARL is an adaptable and versatile framework for cooperative multi-agent learning, so that it can be easily and quickly migrated to tasks with communication settings. This can be achieved by incorporating communication information as part of the input/observation or other customized communication channels.
>
> GoMARL is a generalized framework that does not have any restrictions on communication. Whether agents require similar locality and grouping only relies on your communication settings and the communication mechanism (*e.g.,* environmental constraints, communication objectives, *etc.*). We believe that GoMARL can serve as a very strong foundational framework for researchers in the field of multi-agent communication.
>
> **Q4: Discussion on limitations.**
>
> A4: Thank you for suggesting including a separate subsection of limitations. Here we present our discussion on limitations and future plans: "The automatic grouping mechanism of GoMARL is currently grounded in the value decomposition process, rendering it inapplicable to policy-based methods. Our future research will delve into automatic grouping based on policy gradients to enhance the learning efficiency of policy-based methods. Furthermore, the current version of GoMARL is only designed for single-level grouping, and it possesses the potential to achieve *finer granularity in group division*. We will investigate multi-level grouping (*i.e.,* group-within-a-group) in the future to achieve nested group division, pursuing better performance and higher learning efficiency in more complex environments."
>
> **We would like to express our gratitude once again for your insightful feedback. We hope that our response has addressed your concerns. If there are any remaining uncertainties, we welcome further discussion to ensure clarity and satisfaction.**

---

> > ### Comment · Reviewer_p19m · 2023-08-15
> >
> > Thank you to the authors for clarifications. I have no further questions.

---

> > > ### Author Response · Authors · 2023-08-15
> > > **Thank you**
> > >
> > > We sincerely thank you for taking the time to read our response. We are pleased that our response has effectively addressed your concerns. Thank you once again for your valuable contribution to our submission.

---

### Author Rebuttal · Authors · 2023-08-10

To AC and all the reviewers:

We would like to express our sincere gratitude to AC and all the reviewers for their great efforts in evaluating our paper. We sincerely appreciate the valuable insights and suggestions provided by each of you. We have carefully addressed all the questions and concerns raised in the reviews through our rebuttal comments.

The attached PDF to this global rebuttal comment contains additional experimental results requested by Reviewer y5k7.

If there are any remaining queries or uncertainties after reviewing our responses, we welcome further discussions during the upcoming phase. Your continued engagement is highly valued and appreciated. Thank you once again for your time, expertise, and contribution to our paper.

---

### Decision · Program_Chairs · 2023-09-21

**Decision:**

Accept (poster)

**Comment:**

This paper studies subgrouping of agents for value decomposition-based approaches to cooperative MARL. All reviewers appreciated that the method was novel, clearly explained, and the experimental evaluations were appropriate.